# A Study on the Antibacterial, Antispasmodic, Antipyretic, and Anti-Inflammatory Activity of ZnO Nanoparticles Using Leaf Extract from *Jasminum sambac* (L. Aiton)

**DOI:** 10.3390/molecules29071464

**Published:** 2024-03-25

**Authors:** S. K. Johnsy Sugitha, Raja Venkatesan, R. Gladis Latha, Alexandre A. Vetcher, Bandar Ali Al-Asbahi, Seong-Cheol Kim

**Affiliations:** 1Department of Chemistry, Holy Cross College, Nagercoil, Affiliated to Manonmaniam Sundaranar University, Tirunelveli 627012, India; johnsysugitha@gmail.com; 2School of Chemical Engineering, Yeungnam University, 280 Daehak-Ro, Gyeongsan 38541, Republic of Korea; rajavenki101@gmail.com; 3Department of Chemistry and Research Centre, Holy Cross College, Nagercoil 629002, India; 4Institute of Biochemical Technology and Nanotechnology, Peoples’ Friendship University of Russia n.a. P. Lumumba (RUDN), 6 Miklukho-Maklaya St., 117198 Moscow, Russia; avetcher@gmail.com; 5Department of Physics and Astronomy, College of Science, King Saud University, P.O. Box 2455, Riyadh 11451, Saudi Arabia; balasbahi@ksu.edu.sa

**Keywords:** green synthesis, *Jasminum sambac*, ZnO NPs, antibacterial, antispasmodic, antipyretic, anti-inflammatory activity

## Abstract

The green synthesis of zinc oxide nanoparticles (ZnO NPs) using plants has grown in significance in recent years. ZnO NPs were synthesized in this work via a chemical precipitation method with *Jasminum sambac* (JS) leaf extract serving as a capping agent. These NPs were characterized using UV-vis spectroscopy, FT-IR, XRD, SEM, TEM, TGA, and DTA. The results from UV-vis and FT-IR confirmed the band gap energies (3.37 eV and 3.50 eV) and the presence of the following functional groups: CN, OH, C=O, and NH. A spherical structure and an average grain size of 26 nm were confirmed via XRD. The size and surface morphology of the ZnO NPs were confirmed through the use of SEM analysis. According to the TEM images, the ZnO NPs had an average mean size of 26 nm and were spherical in shape. The TGA curve indicated that the weight loss starts at 100 °C, rising to 900 °C, as a result of the evaporation of water molecules. An exothermic peak was seen during the DTA analysis at 480 °C. Effective antibacterial activity was found at 7.32 ± 0.44 mm in Gram-positive bacteria (*S. aureus*) and at 15.54 ± 0.031 mm in Gram-negative (*E. coli*) bacteria against the ZnO NPs. Antispasmodic activity: the 0.3 mL/mL sample solution demonstrated significant reductions in stimulant effects induced by histamine (at a concentration of 1 µg/mL) by (78.19%), acetylcholine (at a concentration of 1 µM) by (67.57%), and nicotine (at a concentration of 2 µg/mL) by (84.35%). The antipyretic activity was identified using the specific *Shodhan vidhi* method, and their anti-inflammatory properties were effectively evaluated with a denaturation test. A 0.3 mL/mL sample solution demonstrated significant reductions in stimulant effects induced by histamine (at a concentration of 1 µg/mL) by 78.19%, acetylcholine (at a concentration of 1 µM) by 67.57%, and nicotine (at a concentration of 2 µg/mL) by 84.35%. These results underscore the sample solution’s potential as an effective therapeutic agent, showcasing its notable antispasmodic activity. Among the administered doses, the 150 mg/kg sample dose exhibited the most potent antipyretic effects. The anti-inflammatory activity of the synthesized NPs showed a remarkable inhibition percentage of (97.14 ± 0.005) at higher concentrations (250 µg/mL). Furthermore, a cytotoxic effect was noted when the biologically synthesized ZnO NPs were introduced to treated cells.

## 1. Introduction

In recent years, we have seen a rapid development in the area of nanotechnology, with nanoparticles (NPs) emerging as advantageous in the fields of medicine, biology, and other fields [1,2,3]. Nanotechnology is one of the imperative disciplines of research at the moment. Plants and their derivatives are mainly employed in the synthesis of NPs. By virtue of their remarkable biological attributes, NPs have garnered substantial attention and have been used in the areas of drug delivery and medicine, encompassing diagnostics, healthcare, bioimaging, antimicrobial interventions, and the burgeoning field of nanomedicine [4,5]. Nanomedicine, with its interdisciplinary approach, serves as an interlinking conduit, bridging the profound chasm between the intricacies of nanostructures and the complexities of biology. In particular, metallic NPs showcase a myriad of therapeutic potentials, among which ZnO NPs stand out as noteworthy exemplars. Notably, these NPs exhibit exceptional biocompatibility and find widespread utilization across numerous industrial sectors, ranging from environmental sciences, synthetic textiles, and food and packaging to medical care, healthcare infrastructure, construction, and decoration [6,7].

The green synthesis of nanoparticles (NPs) offers several advantages over physical and chemical methods, primarily due to its environmental friendliness, sustainability, and cost-effectiveness. The biological methods are green and safe. There are no negative effects on the environment involved in their production [8]. The uses of ZnO NPs extend far beyond their industrial employment, permeating diverse domains such as drug delivery, material manufacturing, biosensors, microarray technology, tissue engineering, nanoelectronics, energy production, biotechnology, and information technology [9,10,11,12]. Remarkably, these NPs have facilitated significant advancements in the delivery of vaccines and anticancer drugs, thereby contributing to the progress of medical science. The antimicrobial and anti-inflammatory properties inherent to NPs, coupled with their innate biological characteristics, have resulted in a surge in interest within the medicinal field over recent years [13,14]. As a consequence, NPs have emerged as pivotal entities, effectively enhanced the efficacy of therapeutic interventions, and served as potent tools for medical advancement.

ZnO is one among the notable compounds bestowed with the esteemed safety endorsement by the discerning US Food and Drug Administration [15,16]. Considered to be a veritable treasure trove in nanomaterial research, these ZnO NPs emanate a host of commendable biological activities. Their versatile implementation spans a multitude of manufacturing processes, encompassing the realms of rubbers, plastics, ceramics, glass, cement, lubricants, paints, ointments, adhesives, sealants, pigments, foods, flame retardants, and even first-aid tapes [17,18]. The advent of the green synthesis method through the employment of plant extracts has captured considerable attention as a viable and simplistic alternative to conventional physical and chemical approaches [19,20,21,22,23,24]. This eco-friendly methodology boasts inherent advantages such as facile handling and the abundant availability of plant resources, positioning it as a highly desirable avenue for scientific exploration [25]. *Jasminum sambac*, also referred to as Arabian jasmine, belongs to a group of the Oleaceae family, and it is a source of ZnO NPs. JS leaves, replete with flavonoids such as rutin, quercetin, and isoquercetin, boast a rich composition featuring benzyl alcohol, methyl benzoate, and other botanical wonders [26]. As an alternative to other common synthesis methods, it has been shown that the green synthesis of nanoparticles is less hazardous, has a lower cost, and is environmentally friendly [27]. In addition to the earlier indicated characteristics, these ZnO-NPs derived from plant extracts are also known to have excellent antimicrobial properties [28,29,30,31].

The novelty of the present work is the study of the antibacterial, antispasmodic, and antipyretic anti-inflammatory activity of synthesized ZnO NPs utilizing JS leaf extracts. Zinc acetate dihydrate was used as the precursor, and the leaf extract served as the reducing agent. The current investigation is based on the synthesis of ZnO NPs using the renowned JS leaves. The nature of NPs can be explained by characterization techniques like UV-visible spectroscopy, FTIR, XRD, SEM, TEM, TGA, and DTA analysis. Additionally, to improve our understanding of the therapeutic potential of these synthesized NPs, anti-bacterial, anti-spasmodic, and antipyretic, anti-inflammatory activity was also studied.

## 2. Results and Discussion

The JS leaf extracts were visually observed for their color change when incubated with a Zn(CH_3_COO)_2_·2H_2_O solution. The color change of the extract indicated the synthesis of ZnO NPs. The dried sample proceeded with characterization techniques such as FTIR, XRD, UV-vis, SEM, TEM, TGA, DTA, and biological studies. The FTIR, UV-vis, and SEM analysis of the JS leaf extracts are shown in Appendix A, and the phytochemical analysis of these extracts is presented in Appendix A.

### 2.1. FTIR Analysis

The functional groups of the synthesized ZnO NPs were observed via the FTIR spectrum, 100, a PerkinElmer spectrometer with a 4 cm^−1^ resolution in the 4000–400 cm^−1^ range, as shown in Figure 1. The broad spectrum at 3105 cm^−1^ is related to O-H stretching, and the peak seen at 2765 cm^−1^ is due to C-H stretching. The ester group corresponds to the peak at 1760 cm^−1^. The C=C stretching of alkene compounds was observed with the peak at 1550 cm^−1^. The peak measured at 1440 cm^−1^ is suggestive of the bending vibration of CH_3_. The C=Cl stretching of the halo complex is shown by the peak at 825 cm^−1^. Similar results were also observed in various samples of ZnO NPs [32].

### 2.2. XRD Analysis

The crystalline nature was determined via XRD with 2θ varied from 10° to 80°. The crystal plane characteristics are displayed in the X-ray diffractogram. As illustrated in Figure 2, the XRD pattern obtained corresponds to (100), (002), (101), (102), (110), (103), (200), (112), (201), and (202). The purity and high crystalline form of synthesized nanoparticles were confirmed by the lack of observing any other additional peaks. To calculate the particle size, the Debye–Scherrer formula D = kλ/βCosθ was utilized, where θ is the diffraction angle, β is the full width at half maximum (FWHM) = 0.63, and λ is the wavelength of the X-rays utilized (1.54060 Å). It was found that the synthesized ZnO NPs have an average crystallite size of 26 nm. With lattice parameters a = b = 3.2568 Å and C = 5.2125 Å, the NPs were sphere-shaped [33,34].

### 2.3. Results of UV-Visible Spectroscopy Analysis

The absorption spectrum of the obtained nanoparticles shows the optical properties of ZnO NPs. In order to examine the optical absorption of the ZnO nanoparticles synthesized, a UV-visible spectrometer was employed to measure wavelengths ranging from 200 to 800 nm. Figure 3 shows the absorption spectrum of ZnO NPs. The absorption band observed at 355 nm was due to electronic transitions between the valence and conduction bands [35]. According to the UV spectrum of ZnO NPs using JS, the band gap energy was estimated to be 3.49 eV. The difference in the absorption energy was due to the substrate present, which was also responsible for the band gap’s deviation from its real value [36]. Based on these results, the optical and overall efficiency of ZnO is significantly influenced by the substrate utilized. ZnO production and optimization can be used for a variety of applications, including optoelectronics, solar cells, and sensors [37,38]. Its electron–hole confinement in a small volume additionally leads to a quantum size effect, which results in an increased band gap energy value. It can be utilized as a photosensitive material for UV photon detection as a result of its wide band gap [39,40,41,42,43,44,45].

### 2.4. SEM Analysis

The size and surface morphology of the synthesized ZnO NPs were determined with a scanning electron microscope. According to the SEM results, the green synthesized ZnO NPs seem to be spherical [46] in shape, and the size ranges from 20 to 50 nm. The SEM images of ZnO NPs are presented in [Figure 4A–C], where the particle agglomeration takes place due to a change in pH and a decrease in surface energy [47,48,49,50,51,52,53,54]. The EDAX spectrum of the ZnO NPs verifies the presence of zinc and oxygen in proper ratios and impurity-free nanoparticles. Figure 4D illustrates the chemical composition of ZnO nanoparticles, with oxygen comprising 27% and zinc comprising 73% of the elements present.

### 2.5. TEM Analysis

An accelerated voltage of 200 kV was applied to operate the TEM (JEOL, JEM-2100, Tokyo, Japan) to investigate the size and inner morphology of the synthesized ZnO NPs. The size, size distribution, and shape of ZnO NPs mediated JS are shown in Figure 5A,B. According to the TEM images, the ZnO NPs had an average mean size of 26 nm and were spherical in shape, as shown in Figure 5B. Furthermore, ZnO NPs produced by JS are crystalline in nature, as shown by the sequence of rings with bright spots observed in the selected area electron diffraction (SAED) in Figure 5C. The particle size of ZnO NPs measured via the XRD test is in close agreement with the particle size determined by the TEM analysis.

### 2.6. Thermal Analysis

Thermogravimetric and differential thermal analyses were carried out in order to determine the stability and temperature of the ZnO NPs. According to the TGA curve in Figure 6A, weight loss starts at 10 °C to 90 °C as a result of the evaporation of water molecules. The main reason for the weight loss up to 200–400 °C was the decomposition and evaporation of biomolecules and phytochemicals that served as surface capping and stabilizing agents on the ZnO NPs [55]. An exothermic peak was seen during the DTA analysis at 475 °C. In the ZnO phase, there is a powerful exothermic peak between 310 °C and 400 °C in the DTA curve, which represents the burn-out of organic composition Figure 6B. This suggests that the ZnO NPs synthesized via biosynthesis are thermally stable at 400 °C [56].

### 2.7. Results of Biological Studies of ZnO NPs

#### 2.7.1. Antibacterial Activity

The disc diffusion method is utilized to analyze the antibacterial activity of produced ZnO NPs, and the results are represented in Figure 7. Gram-positive (*S. aureus*) and Gram-negative (*E. coli*) bacteria strains were assayed using the disc diffusion method. The ZnO NPs exhibit an increased antibacterial effect against *S. aureus* and *E. coli* bacteria, with the maximum inhibition zone observed against *S. aureus* at 7.32 ± 0.44 mm and *E. coli* at 15.54 ± 0.031 mm. According to differences in the typical characteristics, like composition and structure, these two bacterial strains exhibit distinct inhibitory activities. Compared to *E. coli*, *S. aureus*’s cell wall has a more dense and powerful layer of peptidoglycan and less concentrated negative-charged electrons. If compared to positive-charged bacteria, Gram-negative bacteria, such as *E. coli*, have greater diffusion and electrostatic interactions with ZnO NPs. With regard to particular bacteria, fungi, and viruses, ZnO NPs show a broad-spectrum antibacterial activity that is intended to eliminate, render harmless, or have some sort of regulating effect on pathogenic organisms. The synthesized ZnO NPs displayed excellent antibacterial activity against varieties of bacteria that are medically harmful.

#### 2.7.2. Antispasmodic Activity

In the current investigation, the ZnO NPs underwent a thorough analysis to evaluate their anti-spasmodic activity against involuntary muscle contractions, known as spasms, within the intestinal tract. The study focused on three spasmogens—acetylcholine, histamine, and nicotine—which were chosen for their roles in inducing spasms. By subjecting the biosynthesized ZnO NPs to these spasmogens, the research aimed to unveil their potential as anti-spasmodic agents, shedding light on their capacity to alleviate spasms and relieve associated symptoms. Additionally, the study considered Ayurvedic scriptures, specifically the *Shodhan vidhi* method [57], to enhance the biological activity of guggul and mitigate its toxicity. Guinea pigs were used to examine the antispasmodic activity on the ileum, with the isolated tissue tested against spasmogens. A 0.3 mL/mL sample solution demonstrated significant reductions in stimulant effects induced by histamine (at a concentration of 1 µg/mL) by (78.19%), acetylcholine (at a concentration of 1 µM) by (67.57%), and nicotine (at a concentration of 2 µg/mL) by (84.35%). These findings underscore the sample solution’s potential as an effective therapeutic agent, showcasing its notable antispasmodic activity in alleviating the effects of histamine, acetylcholine, and nicotine on the guinea pig ileum.

#### 2.7.3. Antipyretic Activity

The antipyretic activity was carried out in male Sprague Dawley rats weighing between 180 and 220 g. The antipyretic activity of the ZnO NPs is displayed in Figure 8A. The antipyretic potential of a sample was tested on healthy Sprague Dawley rats induced with pyrexia using a yeast injection. The antipyretic activity of green synthesized ZnO NPs was determined, and the results are shown in Table 1. The experiment involved various groups receiving different doses of the sample, a control group with saline, and a positive control with paracetamol. The rectal temperature of each group was recorded at 1-hour intervals over 5 h. The sample exhibited significant antipyretic activity, statistically reducing hyperthermia in a dose-dependent manner and maintaining effectiveness for up to 5 h after administration.

There was an average rise in rectal temperature following yeast injection, ranging between 2 °C and 2.43 °C, with the inhibition of hyperthermia demonstrating a notable and statistically significant pattern (*p* < 0.001). Among the administered doses, the 100 mg/kg dose of the sample exhibited the most potent antipyretic effect, surpassing even the standard drug paracetamol in its ability to efficiently restore body temperature to normal levels (*p* > 0.05). This result underscores the good efficiency of the sample at the highest dosage, further highlighting its potential as a superior alternative to current antipyretic medications.

#### 2.7.4. Anti-Inflammatory Activity

In the examination of anti-inflammatory activity through the bovine serum albumin (BSA) denaturation assay, meticulous steps were undertaken [58]. Figure 8B shows that the ZnO NPs that were produced using a green synthesis had anti-inflammatory activity. The extent of inhibition of protein denaturation was quantified using a mathematical formula. The calculation involved the determination of the percentage of inhibition achieved, denoted as the inhibition of denaturation (%). The formula employed for this calculation is as follows:$$\mathrm{Inhibitionofdenaturation}=\frac{\mathrm{Acontrol}-\mathrm{Asample}}{\mathrm{Acontrol}}\times 100$$

In this formula, A control represents the absorbance value of the control solution, while A sample represents the absorbance value of the tested compounds. Using this formula, the degree of inhibition of protein denaturation caused by the tested compounds can be precisely determined. This quantitative assessment enables a thorough evaluation of the samples’ effectiveness in maintaining the structural integrity of proteins and, consequently, their potential as agents for combating inflammation. The present research analysis observed the following results regarding the anti-inflammatory activity, such as the percentage of inhibition at various concentrations (µg/mL) of the tested compound.

The data highlight the concentration-dependent effect of the tested compounds on inhibiting protein denaturation. As the concentration of the compounds increased, there was a corresponding rise in the percentage of inhibition observed. At the concentration of 50 µg/mL, the inhibition percentage was measured to be 81.35 ± 0.01, indicating a notable inhibitory effect on protein degradation. Subsequently, as the concentration increased to 100 µg/mL, the percentage of inhibition rose to 87.24 ± 0.005, demonstrating a further enhancement in the compounds’ anti-de-naturation activity. This trend continued with higher concentrations, reaching a remarkable inhibition percentage of 97.14 ± 0.005 at a concentration of 250 µg/mL. These results underscore the dose-dependent nature of the tested compounds, revealing their efficacy in preventing protein denaturation. The data provide valuable insights into the concentration range where the compounds exhibit optimal inhibitory activity, contributing to a better understanding of their potential as anti-inflammatory agents.

#### 2.7.5. Evaluation of Cytotoxicity

According to the study results, ZnO NPs can directly inhibit the activity of the enzyme mitochondrial dehydrogenase, which is in the position of decreasing MTT. It is important to underestimate cellular viability when formazan formation is inhibited by chemotherapeutic chemicals’ effects on active cells that result in cellular cytotoxicity. Here, we used L929 (Fibroblast cells) to study the cell survival of the ZnO NPs quantitatively via MTT assay (Human Colon cancer cells). The results of the 24 h culture of these cells and MTT experiment are shown in Figure 9. The percentage of cell viability and growth inhibition was calculated using the formula below. The formula used to determine the percentage of growth inhibition is as follows:% of viability = Mean OD samples/Mean OD of control group × 100

ZnO NPs seem to be very nontoxic based on the overall cell viability results obtained from L929 cells. ZnO NPs do not cause damage to normal cells with respect to the ZnO NPs concentrations (25 μg/mL, 50 μg/mL, and 100 μg/mL), as shown in Figure 8. When NPs concentrations are at their highest (100 μg/mL), especially if ZnO NPs contents are high, normal cells display some negative effects. The sample of the lowest nanoparticle dosages (5 μg/mL and 10 μg/mL) had minimal effect on L929 cells. ZnO NPs have some cytotoxic against L929 cells. The results of this study indicate that ZnO NPs can activate human fibroblast cells (L929).

## 3. Materials and Methods

### 3.1. Materials

Fresh JS leaves were collected from Kanyakumari, Tamil Nadu, India. The chemicals zinc acetate dihydrate (Zn(CH_3_COO)_2_·2H_2_O and sodium hydroxide (98% purity) used for the analysis were purchased from Merck (Darmstadt, Germany), and all the chemicals were used as received.

### 3.2. Preparation of JS Leaves Extract

In an attempt to eliminate the dust particles, the plant leaves were thoroughly washed with distilled water. After washing, the leaves were permitted to air dry under the shade and powdered [59]. About 10 g of the leaf powder was then properly mixed with 100 mL of deionized water and heated to 70 °C for 30 min. After cooling, whatman No. 1 filter paper was utilized to filter the leaf extract.

### 3.3. Green Synthesis of ZnO Nanoparticles

The green synthesis of ZnO NPs from a solution of JS leaf extract is shown in Figure 10. ZnO NPs were synthesized by mixing 25 mL of JS leaf extract with 10 mL of zinc acetate dihydrate [Zn(CH_3_COO)_2_·2H_2_O] at 65 °C after continuous stirring for 3 h and allowed to withstand for a day. The solution was completely dried and calcined for 2 h at 250 °C, and it gave a fine sample powder, which was stored in airtight bottles for further characterization.

### 3.4. Characterization

#### 3.4.1. Fourier Transform Infrared Spectroscopy (FTIR)

The FTIR SPECTRUM 100, a PerkinElmer (Waltham, MA, USA) spectrometer with a 4 cm^−1^ resolution in the 4000–400 cm^−1^ range, was utilized. KBr was mixed with the sample. An FTIR analysis was carried out on a thin sample pellet that was obtained via pressing it with a hydraulic pellet press [60,61,62].

#### 3.4.2. X-ray Diffraction (XRD)

With 2θ varied from angles 10° to 80°, ZnO NPs, X-ray diffraction patterns were obtained with Cu Kα radiation (λ = 1.54 Å) in an X-ray diffractometer.

#### 3.4.3. UV-Visible Spectroscopy

The absorption spectra of the obtained nanoparticles showed the optical properties of ZnO NPs. A UV-visible spectrometer was used to observe the optical absorption of the synthesized ZnO NPs in the 200–800 nm wavelength range [63].

#### 3.4.4. Scanning Electron Microscopy (SEM)

Using a (TESCAN VEGA3 SBH, Kohoutovice, Czech Republic), scanning electron microscope, the morphology of the ZnO NPs was measured [64]. With the help of scanning electron microscope and an energy-dispersive X-ray analysis (EDAX) instrument, the elemental composition was examined.

#### 3.4.5. Transmission Electron Microscopy (TEM)

We used an accelerated voltage of 200 kV to operate the TEM (JEOL, JEM-2100, Tokyo, Japan) and investigate the size and inner morphology of the synthesized ZnO NPs [65].

#### 3.4.6. Thermal Analysis

Through the use of a PerkinElmer STA-800, the ZnO sample was heated at a rate of 10 °C/min throughout a temperature range of 50 °C to 700 °C for the thermogravimetric analysis (TGA) and differential thermal analysis (DTA).

#### 3.4.7. Biological Studies of ZnO NPs

ZnO NPs’ antibacterial activity against *S. aureus* and *E. coli* was evaluated via the zone of bacterial inhibition method (disc diffusion test) [66,67]. *S. aureus* and *E. coli* were cultured for twenty-four hours in a shaking incubator controlled at 35 °C with constant temperature and humidity. The ZnO NPs samples were placed in a Petri plate and filled with 50 mL of liquid that contained *S. aureus* and *E. coli*. The plates were incubated for 24 h at 35 °C, and then the inhibiting zone values were measured. The antispasmodic activity was determined using the *Shodhan vidhi* method [56]. The antipyretic potential of the ZnO NPs was meticulously assessed using Sprague Dawley rats weighing between 180 and 220 g. In the assessment of anti-inflammatory activity through the Bovine Serum Albumin (BSA) denaturation assay, meticulous steps were taken [68].

#### 3.4.8. MTT Assay

The growth of medium was taken out after an entire day. The concentration of 100 μg/mL was added in triplicate to the relevant wells after being repeatedly diluted multiple times in DMEM using two-fold method (5 μg/mL, 10 μg/mL, 25 μg/mL, 50 μg/mL, and 100 μg/mL of DMEM) [69]. The resultant mixture was then placed in an incubator that had additional humidity and 5% CO_2_ at 37 °C. In addition, control cells that had not been treated were preserved. After a 24 h of treatment, the plate was examined via an Olympus (Tokyo, Japan) CKX41 inversion phase contrast tissue culture microscope equipped with an Optika (Las Vegas, NV, USA) Pro5 CCD camera, and the microscopic results were recorded. Cell morphological modifications, such as rounding or shrinking, granulation, and vacuolization in the cytoplasm, were all taken into consideration as evidence of cytotoxicity [70,71,72].

#### 3.4.9. Statistical Analysis

Three replications of each concentration were used in all studies (antispasmodic, antipyretic, and anti-inflammatory activity). Following a two-way ANOVA analysis of the collected data, Duncan’s multiple-range tests were used to check whether there were significant differences in antispasmodic, antipyretic, and anti-inflammatory activities. The statistical program SPSS-23 (IBM, New York, NY, USA) was utilized for calculating all of the results.

## 4. Conclusions

In conclusion, the cytotoxicity evaluation of the nanoparticles (NPs) revealed critical insights into their biocompatibility and potential applications. ZnO NPs synthesized using JS have ester and hydroxyl groups, which were confirmed by FTIR analysis. XRD studies confirm that the synthesized ZnO NPs have a spherical structure. In the UV-visible spectrum, an absorption band was observed at 355 nm with a band gap energy of 3.49 eV. The SEM images confirm the spherical shape of ZnO NPs. EDAX analysis confirmed the presence of zinc and oxygen elements. TEM images confirmed the size of ZnO NPs as 26 nm. The major weight losses were observed from the TGA, and an exothermic peak was observed at 400 °C. ZnO NPs showed enhanced biological activities. The evaluation of their therapeutic activity revealed significant anti-spasmodic and antipyretic, anti-inflammatory, and antibacterial activity. Effective antibacterial activity was found in Gram-positive (*S. aureus*) and Gram-negative (*E. coli*) bacteria against the green synthesized ZnO NPs. The anti-inflammatory activity of synthesized ZnO NPs showed a remarkable inhibition % of (97.14 ± 0.005) at higher concentrations (250 µg/mL). Overall, the comprehensive cytotoxicity assessment, coupled with biocompatibility evaluation, underscores the necessity of thorough characterization for the safe and effective utilization of NPs in various biomedical and therapeutic contexts. These results propel nanoparticles into the realm of personalized medicine, targeted drug delivery systems, and innovative treatment modalities. As research unravels the intricate mechanisms underlying their therapeutic effects, ZnO NPs hold immense promise for future biological applications.

## Figures and Tables

**Figure 1 molecules-29-01464-f001:**
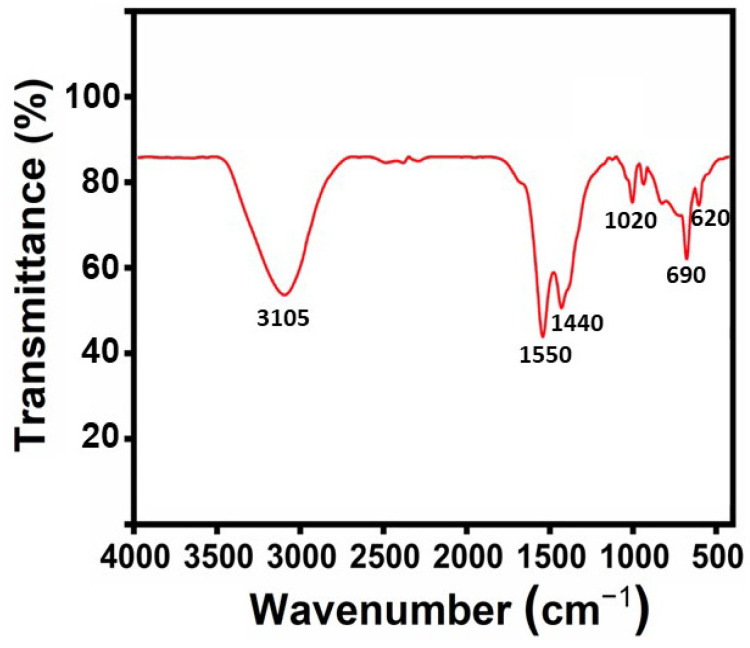
FTIR spectrum green synthesized ZnO NPs.

**Figure 2 molecules-29-01464-f002:**
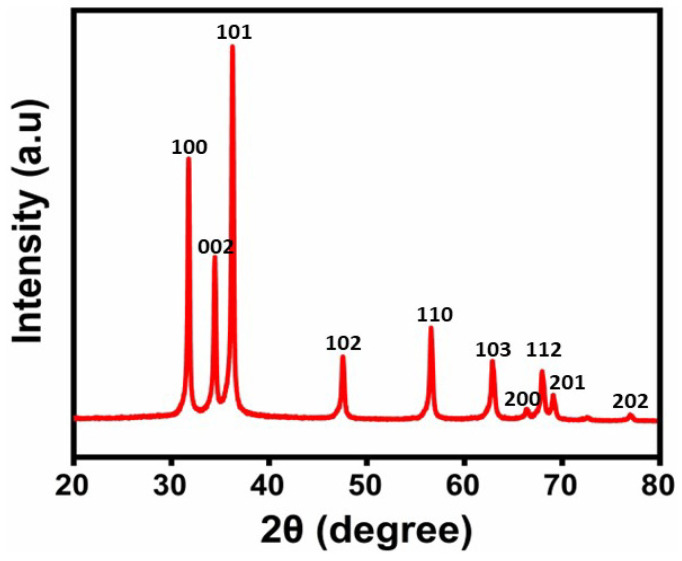
XRD pattern of the green synthesized ZnO NPs.

**Figure 3 molecules-29-01464-f003:**
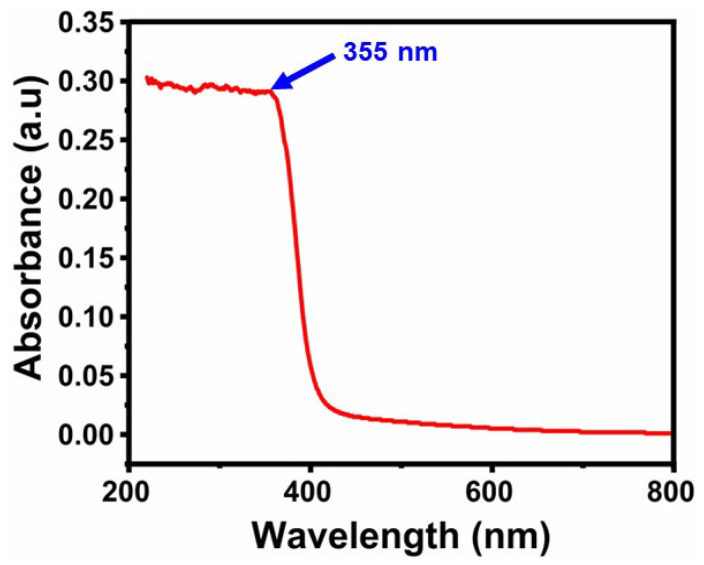
UV-visible absorption spectrum of green synthesized ZnO NPs.

**Figure 4 molecules-29-01464-f004:**
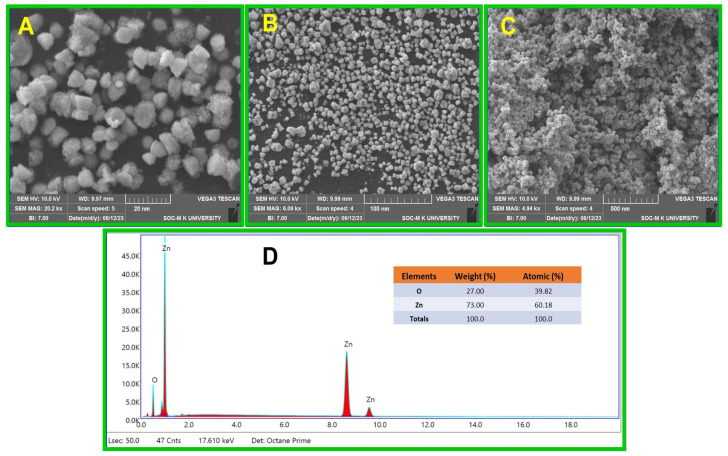
(**A**–**C**) SEM images of ZnO NPs with different magnifications; (**D**) EDAX spectrum of ZnO NPs.

**Figure 5 molecules-29-01464-f005:**
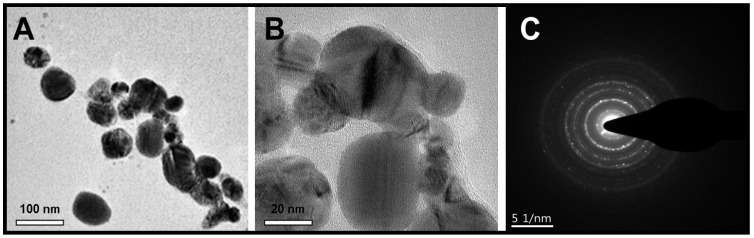
(**A**,**B**) TEM images of ZnO NPs green synthesized by JS; (**C**) SAED pattern of ZnO NPs.

**Figure 6 molecules-29-01464-f006:**
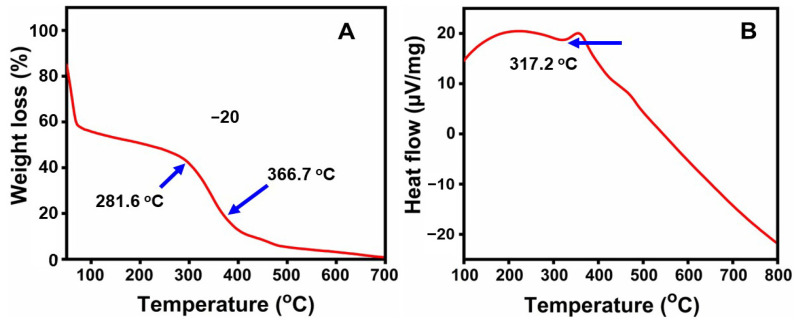
(**A**) TGA curve; (**B**) DTA curve of ZnO nanoparticles.

**Figure 7 molecules-29-01464-f007:**
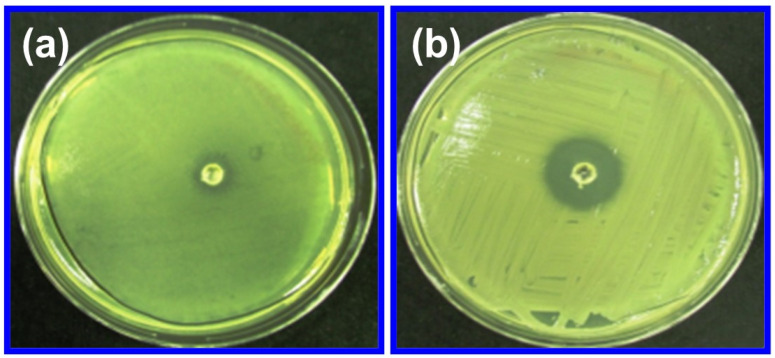
The antibacterial activity of ZnO NPs against (**a**) *S. aureus* and (**b**) *E. coli*.

**Figure 8 molecules-29-01464-f008:**
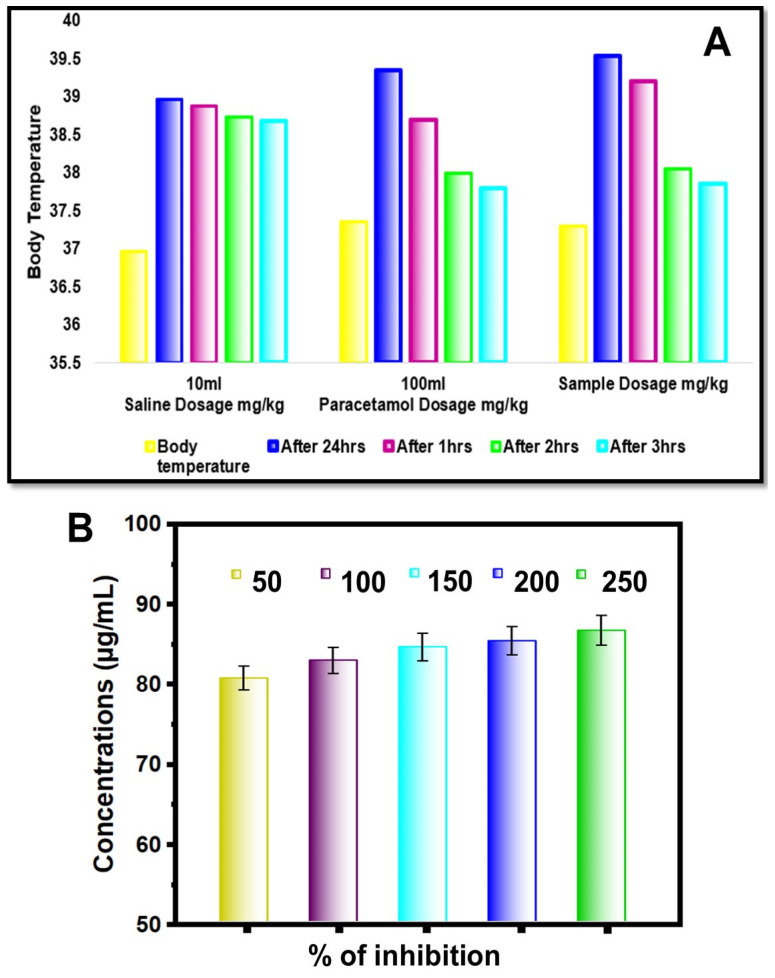
(**A**) Variation chart corresponding to 10 mL of saline, 100 mL of paracetamol, and 100 mL of sample; (**B**) % of inhibition of anti-inflammatory activity.

**Figure 9 molecules-29-01464-f009:**
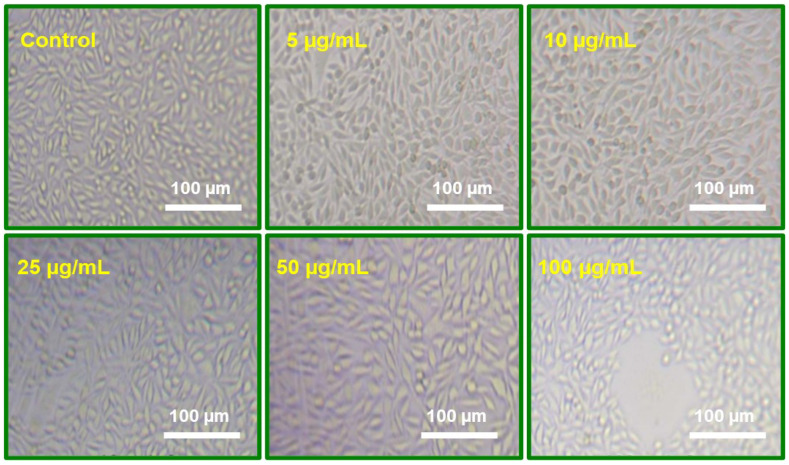
Evaluating the cytotoxicity of ZnO NPs: Images of L929 fibroblast cell lines at different concentrations after a 24 h treatment.

**Figure 10 molecules-29-01464-f010:**
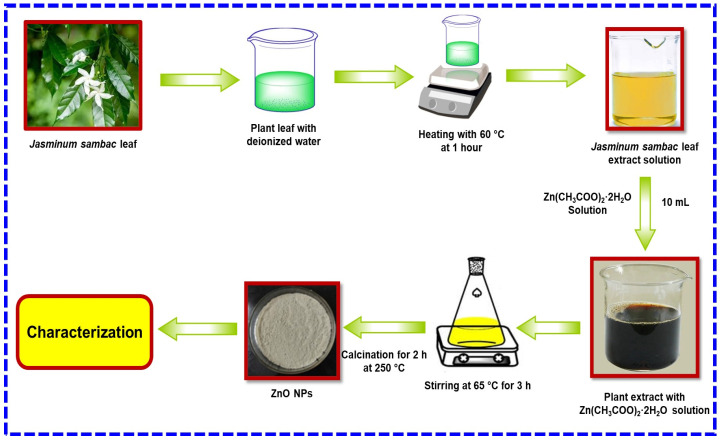
Schematic diagram of synthesis of ZnO NPs using leaf extract of *Jasminum sambac*.

**Table 1 molecules-29-01464-t001:** Antipyretic activity of ZnO NPs with JS leaf extracts.

S. No	Design of Treatment	Dose (mg/kg)	Temperature (°C)	Temperature (°C)	Decrease in Temperature after 2 h (°C)
1.	Saline	10 mL/kg	36.9	38.9	38.7
2.	Paracetamol (standard)	100 mL/kg	36.8	39.0	38.2
3.	Sample (test drug)	100 mL/mg	37.3	39.3	37.9

## Data Availability

Data will be made available on request.

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
