# Peer review of "A Study on the Antibacterial, Antispasmodic, Antipyretic, and Anti-Inflammatory Activity of ZnO Nanoparticles Using Leaf Extract from Jasminum sambac (L. Aiton)"

_molecules, 2024, doi:10.3390/molecules29071464_

Round 1

Reviewer 1 Report

Comments and Suggestions for Authors

Comments to the Author

MS quite well but there are some minor corrections that need to be addressed.

1.         There are several typo and grammar errors that sometimes make the inappropriate understanding of the sentences.

2.         Abstract should contain the significant values obtained in the study which is lacking and should be reframed. Clealry mention the novelty of present work.

3.         Introduction:  The authors should explain the importance of the work in detail in order to attract the readership of this journal.

4. What’s is the advantage of green synthesis of NPs over physical and chemical methods. Should be described in introduction section.  For detail read https://www.degruyter.com/document/doi/10.1515/ntrev-2023-0575/html

5.         Once the binomial nomenclature is used it should be abbreviated thereafter throughout the manuscript.

6. In introduction section pls delete the irrelevant literature. And discuss more about the antimicrobial activity of ZnO synthesized by green methods.

https://www.mdpi.com/2218-273X/10/2/336

7. References needed for 3.4.7. Biological Studies of ZnO NPs and 3.4.8. MTT Assay

8. Authors must add references for SEM, YEM, UV, FTIR, XRD etc techniques. What’s was the nature and size of NPs.

9.  What’s are the phytocompounds present in extract that were responsible for reduction and synthesis of NPs.

10. The cytotoxicity of the NPs should test.

11. Discussion is very poor.

12. Some references are very old. Better to cite last 5 years references.

13. Fig 1 A & C. Authors should provide UV and FTIR spectra of plant extract and compared them with NPs spectra.

14. Similarity index is 27% which is not acceptable. It should be less than 20.

15. What was the standard positive control in antibacterial and MTT assay. Authors should compare their findings with positive control drug. 

Comments on the Quality of English Language

Minor correction 

Author Response

Dear Prof. Dr. Mavis Fu, Editor

We appreciate your kind consideration and constructive comments on our manuscript. As suggested by the editor and reviewers, we have markedly modified the manuscript. The editor’s and reviewer’s questions have been repeated in blue text and our response follows in black. In the revised manuscript, changes are in red.

We would like to thank the Reviewers for thorough reading of this manuscript and for the recommendations that have helped us improve the manuscript quality and scientific value. We hope our revisions have improved the manuscript to a level of the reviewer’s satisfaction.

Sincerely,

Prof. Raja Venkatesan.

Reviewer 2 Report

Comments and Suggestions for Authors

The article titled " Studies on the Antibacterial, Antispasmodic, Antipyretic, and Anti-inflammatory Activity of ZnO Nanoparticles Using Leaf  Extract from Jasminum Sambac" focuses on the synthesis of zinc oxide nanoparticles (ZnO NPs) using a green approach, utilizing Jasminum sambac leaf extract. The study aims to characterize the synthesized nanoparticles and evaluate their potential biomedical applications, particularly in areas such as anti-inflammatory activity and cytotoxicity.

Overall, the manuscript presents a valuable contribution to the field of nanotechnology and green synthesis of nanoparticles. The research question is clearly defined, and the methodology is well-described. However, there are a few minor comments:

-          In line 89 please reformulate “can be explained by the careful applications like FTIR, XRD…”. These are characterization techniques not applications

-          For figures 4A and 4B please write the peaks values on the TGA/DSC curve

-          To enhance readability and visual appeal, consider organizing and separating the images (e.g., Figure 1 A, B, C; Figure 4 A, B), inserting them under each respective characterization technique

-          Line 177 there are some symbols missing for temperature

-          Line 245 in table 1, the word temperature is misspelled

-          Line 306 the figure 7 has the same caption as figure 6

-          The conclusions section could benefit from restructuring to improve coherence and flow. Consider incorporating connection words to enhance the overall structure.

Author Response

Dear Prof. Dr. Mavis Fu, Editor

We appreciate your kind consideration and constructive comments on our manuscript. As suggested by the editor and reviewers, we have markedly modified the manuscript. The editor’s and reviewer’s questions have been repeated in blue text and our response follows in black. In the revised manuscript, changes are in red.

We would like to thank the Reviewers for thorough reading of this manuscript and for the recommendations that have helped us improve the manuscript quality and scientific value. We hope our revisions have improved the manuscript to a level of the reviewer’s satisfaction.

Sincerely,

Prof. Raja Venkatesan.

Reviewer #2:

Comment. 1: The article titled " Studies on the Antibacterial, Antispasmodic, Antipyretic, and Anti-inflammatory Activity of ZnO Nanoparticles Using Leaf  Extract from Jasminum Sambac" focuses on the synthesis of zinc oxide nanoparticles (ZnO NPs) using a green approach, utilizing Jasminum sambac leaf extract. The study aims to characterize the synthesized nanoparticles and evaluate their potential biomedical applications, particularly in areas such as anti-inflammatory activity and cytotoxicity.

Response: We are thankful to the reviewer for insightful comments and suggestions and express gratitude for the time spent reviewing our manuscript. We have modified the manuscript in lieu of the reviewer’s comments.

Comment. 2: Overall, the manuscript presents a valuable contribution to the field of nanotechnology and green synthesis of nanoparticles. The research question is clearly defined, and the methodology is well-described. However, there are a few minor comments:

In line 89 please reformulate “can be explained by the careful applications like FTIR, XRD…”. These are characterization techniques not applications.

Response: The changes that occurred on Line 99 are highlighted in red. NPs can be explained by the characterization techniques like UV-visible spectroscopy, FTIR, XRD, SEM, TEM, TGA, and DTA analysis.

Comment. 3: For figures 4A and 4B please write the peaks values on the TGA/DSC curve.

Response: In response to reviewer comments, we have included the TGA/DTA cure peaks values in Figure 6.

Comment. 4: To enhance readability and visual appeal, consider organizing and separating the images (e.g., Figure 1 A, B, C; Figure 4 A, B), inserting them under each respective characterization technique.

Response: In Figure 1, A, B, and C split in Figures 1, 2, and 3, as per reviewer comments. We have included in the corresponding categorization in Figure 6.

Comment. 5: Line 177 there are some symbols missing for temperature.

Response: In Line 184; the symbol for temperature is presented.

Comment. 6: Line 245 in table 1, the word temperature is misspelled.

Response: In Line 242 in table 1; the word ‘temperature’ has been changed.

Comment. 7: Line 306 the figure 7 has the same caption as figure 6

Response: In Line 311; the caption for Figure 9 has been changed and will be marked in red color.

Comment. 8: The conclusions section could benefit from restructuring to improve coherence and flow. Consider incorporating connection words to enhance the overall structure.

Response: Thank you for your kind comment. The whole article has been revised and checked. In accordance to reviewer comments, we updated a few references and solved all comment. The conclusions section has been restructured according to comments from the reviewer.
